# MiR-15b-5p Expression in the Peripheral Blood: A Potential Diagnostic Biomarker of Autism Spectrum Disorder

**DOI:** 10.3390/brainsci13010027

**Published:** 2022-12-22

**Authors:** Rie Hosokawa, Yuta Yoshino, Yu Funahashi, Fumie Horiuchi, Jun-ichi Iga, Shu-ichi Ueno

**Affiliations:** 1Department of Neuropsychiatry, Molecules and Function, Ehime University Graduate School of Medicine, Shitsukawa, Toon 791-0295, Ehime, Japan; 2Center for Child Health, Behavior and Development, Ehime University Hospital, Toon 791-0295, Ehime, Japan

**Keywords:** autism spectrum disorder, diagnostic biomarker, miRNA, TGF-β

## Abstract

Background: Autism spectrum disorder (ASD), is a neurodevelopmental disorder that is known to have a high degree of heritability. Diagnosis of ASD is difficult because of the high heterogeneity of the clinical symptoms. MicroRNAs (miRNAs) can potentially be diagnostic biomarkers for ASD, and several studies have shown the relationship between miRNAs and ASD pathogenesis. In this study, we investigated ten miRNA and mRNA expression of target genes in peripheral blood to explore a diagnostic biomarker for ASD. Methods: We recruited control and ASD subjects for the discovery cohort (*n* = 6, each) and replication cohort (*n* = 20, each). Using qPCR, miRNA and mRNA expression was measured using the SYBR green and probe methods, respectively. In-silico prediction was used for identifying target genes of miRNAs. An in vitro experiment using HEK293 cells was conducted to investigate whether miR-15b-5p modulates the predicted target genes (TGFBR3 and MYBL1). Results: miR-15b-5p expression indicated an increased trend in the discovery cohort (*p* = 0.052) and a significant upregulation in the replication cohort (*p* = 0.021). In-silico analysis revealed that miR-15b-5p is relevant to cell development and Wnt signaling. The decreased trends of TGFBR3 and MYBL expression were the same as in previous RNA-seq data. MiR-15b-5p positively regulated TGFBR3 expression in in vitro experiments. Conclusions: Upregulated miR-15b-5p expression may represent a useful diagnostic marker of ASD subjects, and it may regulate TGFBR3 mRNA expression. These findings indicate a new perspective in the understanding of the pathogenesis of ASD.

## 1. Introduction

Autism spectrum disorder (ASD) is a neurodevelopmental disorder featuring persistent deficits in social interaction and restricted repetitive behaviors. The prevalence of ASD is higher in males than females, and the male to female ratio in ASD is approximately 4.2:1 [1]. The global prevalence of ASD is 1% [2]; however, the number of children diagnosed with ASD has increased [3,4]. ASD is a lifelong disorder from infancy to adulthood with early altered brain development [5,6]. The need for therapies and support depends on the cognitive ability, independence, and demands of daily life in the individual throughout a lifetime [7]. ASD is under a high degree of heritable control [8]. Among many etiological factors responsible for the onset of ASD, the genetic component is around 50–60% [9]. A postmortem study has suggested that smaller cells and the cell packing density increase in the limbic system, hippocampus, amygdala, and entorhinal cortex, and the number of Purkinje cells also reduces in the cerebellum [6]. As for the risk of ASD, antenatal factors including metabolic disorders of mothers, weight gain, and hospital admission by bacterial or viral infections are common [10]. Specifically, the aging of parents (maternal age: ≥40 years, paternal age: ≥50 years, respectively) are known to associate with the incidence of ASD [10].

Although symptoms of ASD appear in infancy, the average age diagnosed with ASD is approximately 4–5 years old [11]. According to the National Institute for Health and Clinical Excellence guidelines concerning the diagnosis of ASD, it is recommended to use a formal assessment tool, such as the Autism Diagnostic Observation Schedule Second Edition (ADOS-2) or Autism Diagnostic Interview-Revised (ADI-R) [12,13]. However, the high diversity of the clinical symptoms in ASD makes it unclear and difficult to diagnose, particularly in early childhood [11]. Children without the delay in language development, or who are females, often receive later diagnoses [14]. The diagnosis of ASD is based on clinical behaviors because there are no dependable biomarkers [10]. Exploring the reliable diagnostic biomarkers of ASD is necessary for clinical situations and reliable research.

MicroRNAs (miRNAs) are very short (18–25 nucleotides), single-stranded non-coding RNAs [1]. miRNAs regulate approximately 2/3 of human mRNAs [15] and usually bind to the 3′-untranslated region (UTR) of the target mRNAs to repress protein synthesis [16]. miRNAs can circulate within exosomes or RNA-binding proteins and travel extracellularly to modify gene expression in distant tissues [17]. Therefore, the potential of measuring miRNAs as diagnostic or treatment biomarkers in body fluids has been attracting attention, and several studies have reported that miRNA expression in the peripheral blood has the potential to be a biomarker in psychiatric disorders [18,19,20]. Several studies on autopsy brain tissue from ASD patients identified 91 miRNAs that were differentially expressed compared to typically developing individuals [21]. Additionally, in the peripheral blood, a number of miRNAs were found to be relevant to ASD [22,23]. However, further study is necessary for developing peripheral miRNAs into reliable diagnostic biomarkers.

Given this background for exploring peripheral biomarkers, the present study aimed to investigate (1) ten miRNA expressions in peripheral blood using the discovery cohort, (2) the predicted target expression of miR-15b-5p in the discovery cohort, (3) whether miR-15b-5p regulates the predicted target genes, and (4) ten miRNA and target gene expressions in peripheral blood using the replication cohort.

## 2. Methods

### 2.1. Patients with ASD and Controls

Six controls with neurotypical development and six ASD subjects used in previous reports (Table 1, 33931079), were set as the discovery cohort. Next, 20 controls with neurotypical development and 20 ASD subjects were recruited for the replication cohort. The demographic and clinical data of the replication cohort are presented in Table 2. Although we have used two cohorts, this study is a pilot study due to the small number of subjects. The diagnosis of ASD was established using the Diagnostic and Statistical Manual of Mental Disorders, Fifth Edition (DSM-5). All candidate patients were diagnosed with ASD by at least two expert psychiatrists through detailed medical examinations and medical record check-ups. Similar to the ASD subjects, according to the medical examinations, control subjects were verified as neurotypical and psychologically healthy without psychiatric symptoms and an episode of psychiatric disorders or developmental concerns. The control and ASD subjects were Japanese and had no blood relationship. This study was approved by the ethical review boards of Ehime University Graduate School of Medicine (Approval Number: 31-K8). The informed consents were obtained for all participants before the acquisition of blood samples. Intelligence Quotient (IQ) was measured by the Wechsler Adult Intelligence Scale—Third Edition, Wechsler Adult Intelligence Scale—Fourth Edition, Wechsler Intelligence Scale for Children—Third Edition, Tanaka-Binet test, or the Japanese Adult Reading Test.

### 2.2. Collection of Blood Samples, and RNA Isolation

Total RNA was extracted from whole peripheral blood samples using PaxGene Blood RNA Systems tubes (Becton, Dickinson and Company, Tokyo, Japan). RNA was then isolated using the manufacturer’s protocols. The quality and concentration of RNA were quantified using the NanoDrop-1000 system (Thermo Fisher Scientific, Yokohama, Japan). The acceptable standard for RNA samples was a 260/280 ratio between 1.8 and 2.0. Total RNA was stored at −80 °C until use.

### 2.3. Synthesis of Complementary DNA (cDNA) for miRNA-Specific Gene Expression and Quantitative PCR (qPCR)

The Mir-X miRNA First-Strand Synthesis Kit (Takara Bio Inc., Tokyo, Japan) was used to conduct the reverse transcription (RT). The cDNA for miRNA was synthesized from a 5 μL reaction mixture containing RNA (0.25–8.0 μg; 1.875 μL). Then, we added water to the cDNA for miRNA to dilute the mixture to 1:10. The cDNA for miRNA was amplified using a StepOnePlus Real-Time PCR System (Applied Biosystems) with the Mir-X miRNA qRT-PCR TB Green Kit (Takara Bio Inc., Tokyo, Japan) and miRNA-specific primers to measure the miRNA expression levels in duplicate. U6 in the Mir-X miRNA qRT-PCR TB Green Kit was used as an internal standard. RT-qPCR was performed in a final volume of 11.5 µL with 1.0 μL cDNA. In line with the manufacturer’s protocol, the thermal cycling conditions were as follows: initial denaturation step (10 s at 95 °C), 40 cycles of denaturation steps (5 s at 95 °C and annealing and elongation for 20 s at 60 °C), and dissociation curve step (60 s at 95 °C, 30 s at 55 °C, and 30 s at 95 °C). The qPCR forward primer sequences are shown in Appendix A. The expression levels of miRNA were analyzed using Livak’s ΔΔCt method [22].

### 2.4. Synthesis of cDNA- or mRNA-Specific Gene Expression and qPCR

The High Capacity cDNA Reverse Transcription Kit (Applied Biosystems, Foster City, CA, USA) was used for the reverse transcription reaction. The total reaction volume of the reaction mixture containing RNA (0.5 μg per sample) was 40 μL. The mRNA expression levels were quantified in duplicate using a StepOnePlus Real-Time PCR System with the PrimeTime Gene Expression Master Mix (Integrated DNA Technologies, Inc., Coralville, IA, USA) and mRNA-specific probes. We used Hs.PT.58.4282790 for TGFBR3 and Hs.PT.58.3146944 for MYBL1, as mRNA-specific probes. Hs.PT.39a.22214836 for GAPDH was used as an internal standard. RT-qPCR was performed in 10 µL mixtures with 0.5 μL cDNA. According to the manufacturer’s protocol, the thermal cycling conditions were as follows: initial denaturation step (2 min at 50 °C, and 10 min at 95 °C), and 40 cycles of denaturation steps (15 s at 95 °C and annealing and elongation for 60 s at 60 °C). The expression levels of mRNA were calculated using Livak’s ΔΔCt method [24].

### 2.5. Target Gene Prediction of miRNA and Functional Annotation of Predicted Target Genes

miRDB (http://mirdb.org/, accessed on 15 December 2022) was used for target gene prediction, and a target score of 90 or more was set as a reliable score. Subsequently, predicted target genes were subjected to the ClueGO plugin in the Cytoscape program [25] to perform functional annotation of biological processes (BPs). Statistical significance was set at *p* < 0.05 with the post hoc Benjamini–Hochberg method. Based on significant BP terms, the graphical network was generated according to the following criteria: visual style = groups; GO term/pathway network connectivity = medium (kappa score = 0.40).

### 2.6. In Vitro Cell-Line-Based Study

HEK293 cells were cultured in Dulbecco′s Modified Eagle′s Medium/Nutrient Mixture F-12 Ham (DMEM/F-12, Thermo Fisher Scientific) containing 10% fetal bovine serum with penicillin-streptomycin (10,000 U/mL, Thermo Fisher Scientific). The cells were incubated at 37 °C in a 5% CO2 atmosphere. miR-15b-5p (miR-15b-5p mimic (C-300587-05-0002); Dharmacon GE Life Sciences, Lafayette, CO, USA) oligos were transfected into HEK293 cells using Lipofectamine RNAiMAX (Invitrogen, Grand Island, NY, USA) and harvested 48 h post-transfection for target gene expression analysis. RNAs were isolated using Trizol Reagent (Invitrogen).

### 2.7. Statistical Analysis

SPSS 27.0 software (IBM Japan, Tokyo, Japan) was used for the statistical analysis. The assessment of normal distribution was performed using the Shapiro–Wilk test. The homoscedasticity was assessed by using the Levene’s test. The average differences in age and mRNA or miRNA expressions were investigated by the Student’s *t*-test, the Welch’s *t*-test, or the Mann–Whitney U test. Sex differences in control and ASD subjects were compared using Fisher’s exact test. The correlation of demographic data (age, gender, and IQ) with miRNA and mRNA expressions was investigated by the Pearson correlation coefficient or Spearman’s rank correlation coefficient. Statistical significance was defined at the 95% level (*p* = 0.05).

## 3. Results

### 3.1. Demographic Data of ASD Subjects and Controls

The detailed demographic data of the discovery cohort are shown in the previous report (Table 1) [26]. The ratio of males to females was 13:7 in the replication cohort. No significant differences were found in the demographic data between the controls and ASD subjects in the replication cohort (age: *p* = 0.732; sex: *p* = 1.0; Table 2). The average IQ of the total 14 ASD subjects among discovery and replication cohorts was 88.6 ± 24.1 (average ± S.D.).

### 3.2. miRNA Expression in Peripheral Blood of Discovery Cohort

Based on previous reports [19], we decided to examine all ten miRNAs that had altered expression levels both in the peripheral blood and central nervous system (CNS) among ASD subjects. Of those miRNA expressions, the miR-15b-5p expression level was higher in ASD subjects than in the controls, but there were no significant differences (control vs. ASD = 1.00 ± 0.41 vs. 1.93 ± 0.80, *p*  =  0.051; Figure 1A). There were no significant differences in other miRNAs’ expression levels: miR-15a-5p (*p* = 0.485), miR-19b-3p (*p* = 0.485), miR-27a-3p (*p* = 0.643), miR-106b-5p (*p* = 0.485), miR-320-5p (*p* = 0.212), miR-320a-3p (*p* = 0.864), miR-451a (*p* = 0.338), miR-494-5p (*p* = 0.240), miR-494-3p (*p* = 0.607) (Figure 1B–J).

### 3.3. Target Gene Prediction of miR-15b-5p and Validation qPCR in the Discovery Cohort

As a result of the target gene prediction concerning miR-15b-5p, 289 genes passed our criteria (Appendix A). When subjecting those 289 genes to functional annotation, cell development, including axonogenesis and the Wnt signaling pathway was found (Figure 2 and Appendix A). When considering the upregulation of miR-15b-5p expression, we selected the two genes that were significantly downregulated in the RNA-seq data of the discovery cohort: Transforming Growth Factor Beta Receptor 3 (TGFBR3) and MYB Proto-Oncogene Like 1 (MYBL1). TGFBR3 and MYBL1 mRNA expression levels decreased in the ASD group compared with the controls but did not reach a significant level (*p* = 0.076 and *p* = 0.094, respectively; Figure 3). When compared to the previous RNA-seq data, those decreased trends were the same (TGFBR3: 0.59-fold change, *p* = 0.001, q value = 0.0077; MYBL1: 0.54-fold change, *p* = 0.00005, q value = 0.0048) [26].

### 3.4. In Vitro Cell-Line-Based Studies

In the miR-15b-5p oligo-transfected HEK293 cells, about 1,000 times upregulation of miR-15b-5p expression was found (vehicle vs. mimic = 1.00 ± 0.08 vs. 1020.95 ± 461.28, *p* = 0.004; Appendix A). Significant upregulation was found for TGFBR3 expression (vehicle vs. mimic = 1.00 ± 0.07 vs. 1.40 ± 0.13, *p* < 0.001; Figure 4A). No significant changes were observed in MYBL1 (*p* = 0.412; Figure 4B).

### 3.5. miRNA and mRNA Expression in Peripheral Blood of Replication Cohort

In the expression of miRNA, the miR-15b-5p expression level was significantly higher in ASD subjects than in the control subjects (control vs. ASD = 1.00 ± 1.08 vs. 1.48 ± 1.18, *p* = 0.021; Figure 5A). No significant differences were found in other miRNAs’ expression levels: miR-15a-5p (*p* = 0.602), miR-19b-3p (*p* = 0.841), miR-27a-3p (*p* = 0.678), miR-106b-5p (*p* = 0.947), miR-320-5p (*p* = 0.369), miR-320a-3p (*p* = 0.183), miR-451a (*p* = 0.758), miR-494-5p (*p* = 0.841), miR-494-3p (*p* = 0.758) (Figure 5B–J). Moreover, we investigated the TGFBR3 and MYBL1 mRNA expression levels, and no significant differences were observed between the control and ASD subjects (*p* = 0.738 and *p* = 0.994, respectively; Figure 6).

### 3.6. Correlation of miRNA and mRNA Expression with Demographic Data

A significant correlation between gender and miR-15-5p was found both in the control (*p* = 0.030) and ASD subjects (*p* = 0.003). IQ was significantly associated with TGFBR3 (*p* = 0.023) and MYBL1 (*p* = 0.036). Other than those, there were no significant changes (Appendix A). Additionally, the average expression of miR-15b-5p was compared between control and ASD subjects using a linear regression model, including gender as a covariate. A significant trend in the discovery cohort (*p* = 0.052, standardized beta coefficient = 0.593) and a significant result in the replication cohort (*p* = 0.007, standardized beta coefficient = 0.413) were found.

## 4. Discussion

We clarified whether ten miRNAs in the peripheral blood are suitable diagnostic biomarkers using discovery and replication cohorts and investigated the target genes of miR-15b-5p (candidate miRNA biomarker). Upregulated miR-15b-5p was confirmed in both cohorts, and may represent a useful diagnostic biomarker of ASD. In the in-silico prediction, miR-15b-5p was possibly associated with cell development including axonogenesis and the Wnt signaling pathway. Moreover, TGFBR3 was positively regulated by miR-15b-5p as a result of the in vitro experiment.

miR-15b-5p is relevant to various pathogeneses such as different types of cancers, coronary artery disease, diabetic complications, and dyslipidemia [27]. Among neuropsychiatric diseases, in vitro studies have shown that miR-15b-5p is relevant to the pathogenesis of Alzheimer’s disease [28] and Parkinson’s disease [29]. Abu-Elneel et al. (2008) reported that the miR-15b expression level was increased in the cerebellar cortex of ASD subjects [30]. Based on our in-silico study, miR-15b-5p is associated with cell development including axonogenesis and the Wnt signaling pathway via the regulation of target genes. Actually, previous reports have shown associations of ASD pathogenesis with axon guidance and myelination [31] and Wnt signaling [32]. In terms of peripheral miR-15b-5p expression, one study reported downregulated expression in ASD subjects [33]. Inconsistent with that report, upregulated expression of miR-15b-p was found in the present study. miRNA expression possibly changes as a result of life-course environmental factors [34] as well as disease pathogenesis and severity. Gender affected the miR-15b-5p expression in the present study. When applying the miR-15b-5p expression change to the clinical situation as a diagnostic marker, we should consider the other factors which affect its expression.

Due to the trends of upregulated miR-15b-5p and downregulated TGFBR3 in the discovery cohort and in-silico prediction, we presumed that miR-15b-5p inversely regulates TGFBR3 and MYBL1 expression. However, upregulated TGFBR3 was confirmed in the miR-15b-5p-overexpressing HEK293 cells. Although miRNAs generally work as a suppressor of mRNA expression, miRNA-mediated upregulation of mRNA expression was also reported [35], and some miRNAs’ up- or downregulation depends on specific conditions and factors [36,37]. In the future, mechanistic experiments such as dual-luciferase assays are necessary to elucidate whether miR-15b-5p regulates TGFBR3 mRNA expression by binding the seed sequence of the 3′-UTR of the TGFBR3 gene.

TGFBR3 is a membrane proteoglycan that works as a co-receptor with other TGF-β receptor superfamily members. Decreased TGFBR3 mRNA expression is consistent with the decreased TGF-β in the plasma reported in a previous meta-analysis [38]. At the functional level, TGF-β signaling is relevant to a regulator of interneuron neurogenesis [39] and autophagy in ASD [40]. Taken together, miR-15b-5p possibly plays a key role in ASD pathogenesis by regulating the expression of TGFBR3, which has potential roles both in the brain and peripheral tissues.

In the replication cohort, successful replication was found in miR-15b-5p expression but not in TGFBR3 mRNA expression. Recently, the correlation between ASD severity and immune dysfunction, especially cytokines, has been confirmed [41,42]. However, the severity of ASD symptoms was assessed in the discovery cohort but not in the replication cohort. To reach a conclusion at this point, a further study should be conducted with a large sample of ASD subjects, including an assessment of symptoms.

There were four limitations in this study. First, the sample size was still relatively small even though we prepared discovery and replication cohorts. Second, we focused on only adults with ASD. Typically, ASD is seen as an innate condition [43], and miR-15 is known to express in the umbilical cord blood [44]. Third, the severity of ASD was not considered when measuring miRNA and mRNA expressions. Finally, expression of ten miRNAs were not corrected by multiple comparisons due to the small sample size and variance of the expression. In the future, a replication study with a large number of samples including children and adults with ASD with clinical severity should be conducted.

## 5. Conclusions

Upregulated miR-15b-5p expression may represent a useful diagnostic biomarker of ASD subjects, and it possibly regulates TGFBR3 mRNA expression. These findings suggest a new perspective in understanding the pathogenesis of ASD.

## Figures and Tables

**Figure 1 brainsci-13-00027-f001:**
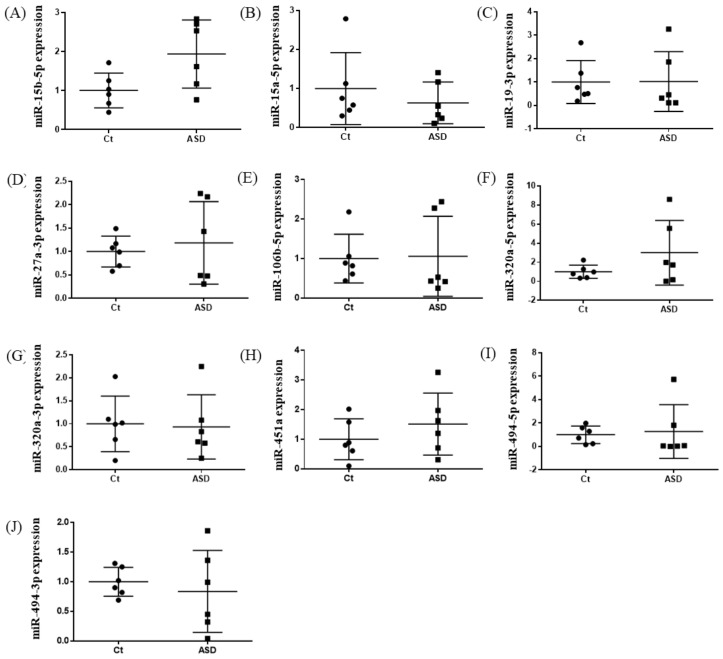
Expression of ten miRNA in the discovery cohort. The *y*-axis represents the ratio of relative expression values of the control and ASD subjects. The expression of ten miRNAs were measured by qPCR: (**A**) miR-15b-5p, (**B**) miR-15a-5p, (**C**) miR-19b-3p, (**D**) miR-27a-3p, (**E**) miR-160b-5p, (**F**) miR-320a-5p, (**G**) miR-320a-3p, (**H**) miR-451a, (**I**) miR-494-5p, and (**J**) miR-494-3p. miR, microRNA; Ct, controls; ASD, autism spectrum disorder.

**Figure 2 brainsci-13-00027-f002:**
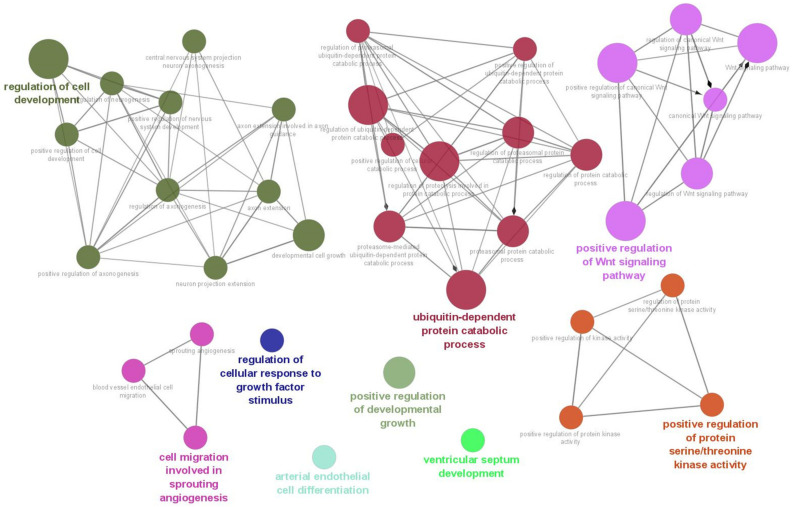
Biological process result based on target genes of miR-15b-5p. The network was constructed with whole biological process terms based on 289 target genes of miR-15b-5p using ClueGo in the Cytoscape program.

**Figure 3 brainsci-13-00027-f003:**
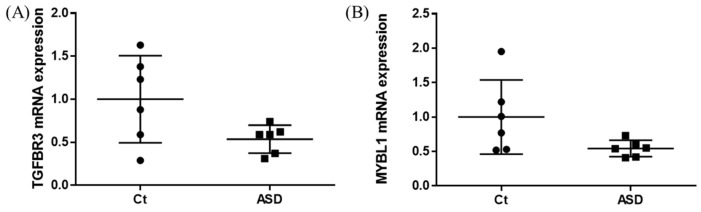
TGFBR3 and MYBL1 mRNA expression in the discovery cohort. The *y*-axis represents the ratio of the relative expression value of the control and ASD subjects for TGFBR3 (**A**) and MYBL (**B**). Ct, controls; ASD, autism spectrum disorder.

**Figure 4 brainsci-13-00027-f004:**
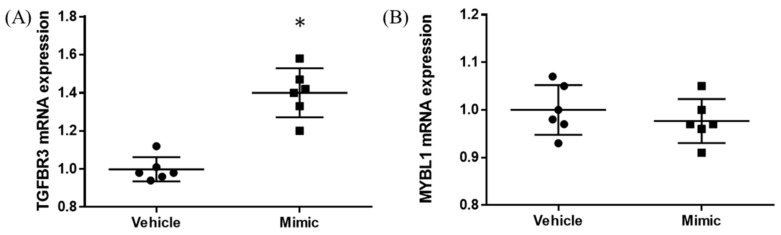
TGFBR3 and MYBL1 mRNA expression results from in vitro study. The *y*-axis represents the ratio of the relative expression value of the TGFBR3 (**A**) and MYBL1 (**B**). * *p* < 0.05.

**Figure 5 brainsci-13-00027-f005:**
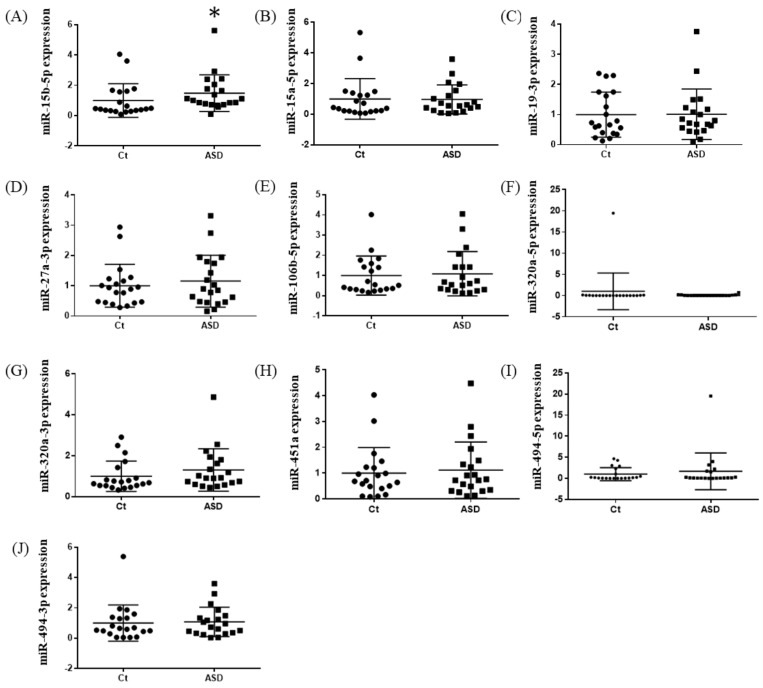
Expression of ten miRNAs in the replication cohort. The *y*-axis represents the ratio of relative expression values of the control and ASD subjects. The expression of the ten miRNAs were measured by qPCR: (**A**) miR-15b-5p, (**B**) miR-15a-5p, (**C**) miR-19b-3p, (**D**) miR-27a-3p, (**E**) miR-160b-5p, (**F**) miR-320a-5p, (**G**) miR-320a-3p, (**H**) miR-451a, (**I**) miR-494-5p, and (**J**) miR-494-3p. miR, microRNA; Ct, controls; ASD, autism spectrum disorder. * *p* < 0.05.

**Figure 6 brainsci-13-00027-f006:**
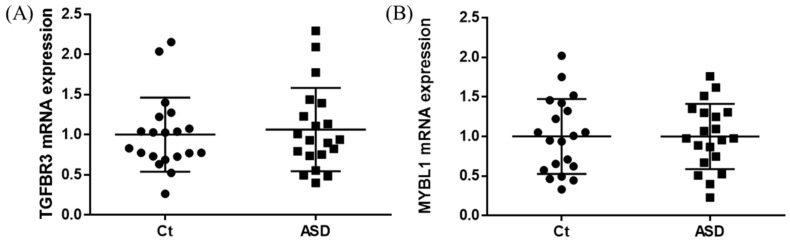
TGFBR3 and MYBL1 mRNA expression in replication cohort. The *y*-axis represents the ratio of the relative expression value of the control and ASD subjects for TGFBR3 (**A**) and MYBL (**B**). Ct, controls; ASD, autism spectrum disorder.

**Table 1 brainsci-13-00027-t001:** Demographic and clinical data of discovery cohort.

	Ct	ASD	*p* Value
Number of samples	6	6	
Age (years mean ± S.D.)	30.2 ± 8.5	31.7 ± 8.5	1.0
Male:Female	5:1	5:1	1.0

Values denote mean ± standard deviation; Ct: Control, ASD: Autism Spectrum Disorder.

**Table 2 brainsci-13-00027-t002:** Demographic and clinical data of replication cohort.

	Ct	ASD	*p* Value
Number of samples	20	20	
Age (years mean ± S.D.)	30.9 ± 8.3	30.0 ± 8.2	0.732
Male:Female	13:7	13:7	1.0

Values denote mean ± standard deviation: Ct: Control, ASD: Autism Spectrum Disorder.

## Data Availability

The datasets used and/or analyzed during the current study are available from the corresponding author on reasonable request.

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
