# Peer review of "MiR-15b-5p Expression in the Peripheral Blood: A Potential Diagnostic Biomarker of Autism Spectrum Disorder"

_brainsci, 2022, doi:10.3390/brainsci13010027_

Round 1
Reviewer 1 Report (Previous Reviewer 2)
Re-review
As a reviewer, I carefully read the revised version of the article. Changes and additions made by the authors raised the scientific value of the article to a higher level.
Nevertheless, I urge the authors to clearly state that their study is a pilot study due to existing limitations, e.g. small number of the subjects. This can be mentioned in the title of the article or in the "Methods" section.
Author Response
As a reviewer, I carefully read the revised version of the article. Changes and additions made by the authors raised the scientific value of the article to a higher level.
Nevertheless, I urge the authors to clearly state that their study is a pilot study due to existing limitations, e.g. small number of the subjects. This can be mentioned in the title of the article or in the "Methods" section.
Response: Thank you for the critical comment. Due to the small number of subjects, we also think that this study is a pilot study, and further investigation is necessary to elucidate whether this miRNA is a reliable biomarker.
We have added a sentence to clarify this point (line 83) as below.
Although we have used two cohorts, this study is a pilot study due to the small number of subjects.
Reviewer 2 Report (Previous Reviewer 1)
The study examined miRNA expression in peripheral blood using a discovery cohort, the predicted target expression of miR-15-5p in the discovery cohort, whether miR-15b-5p regulates the predicted target genes, and 10 miRNA and target 76 gene expressions in peripheral blood using the replication cohort.
The topic of peripheral markers to examine ASD is relevant and interesting. The study is an important follow-up to a published report from 2014. The manuscript is an important follow-up to the first report of serum microRNA in ASD (Vasu et al., 2014). The manuscript is very well written. The text is clear and easy to read. The conclusions are consistent with the evidence provided in the manuscript. The main question - whether microRNAs may serve as a biomarker for autism spectrum disorder - is clearly addressed.
Author Response
The study examined miRNA expression in peripheral blood using a discovery cohort, the predicted target expression of miR-15-5p in the discovery cohort, whether miR-15b-5p regulates the predicted target genes, and 10 miRNA and target 76 gene expressions in peripheral blood using the replication cohort.
The topic of peripheral markers to examine ASD is relevant and interesting. The study is an important follow-up to a published report from 2014. The manuscript is an important follow-up to the first report of serum microRNA in ASD (Vasu et al., 2014). The manuscript is very well written. The text is clear and easy to read. The conclusions are consistent with the evidence provided in the manuscript. The main question - whether microRNAs may serve as a biomarker for autism spectrum disorder - is clearly addressed.
Response: Thank you for the great comment. We believe that these studies including the first report of serum microRNA in ASD (Vasu et al., 2014) will be useful to make reliable biomarkers that can be used in the clinical situation.
This manuscript is a resubmission of an earlier submission. The following is a list of the peer review reports and author responses from that submission.
Round 1
Reviewer 1 Report
The authors have produced a follow up study to their publication (Horiuchi et al., 2021) on the same discovery and replication set of subjects with autism spectrum disorder (ASD) and matched control subjects. Here they examine microRNA from whole peripheral blood samples that may regulate gene expression changes noted in the prior study. They have focused on miR-15b-5p and predicted gene targets TGFBR2 and MYBL1.
Minor point:
There is a typo in Table 1 regarding the age of control subjects (should likely be 30.17 and not 301.7).
In Figures 1 and 5, the correct y-axis label for C) should read miR-19b-3p to be in agreement with the figure legends.
Major issues:
There is a widespread range in miR-15b-5p expression in the ASD subjects in Figure 1A. Examining the discovery ASD cohort as outlined in the Horiuchi et al. (2021) manuscript reveals significant clinical divergence with subjects 3-6 having significantly lower IQs, and therefore lower level of overall functioning, than subjects 1-2. Subjects 3-6 are profoundly impaired, whereas the first two are average. This significant clinical difference may contribute to the expression variance in Figure 1.
Although the authors note the sample size as a limitation, the low number of subjects in both the discovery and replication cohorts make interpretation of the results questionable. Significantly increasing the cohort sizes and correcting for overall clinical functioning may yield data demonstrating statistically significant differences in miR expression between the cohorts. Additionally, the lack of data on severity of ASD symptoms in the replication cohort is another complication that makes interpreting the data difficult.
The statistical comparisons involving data on miR expression in Figures 1 and 5 should be corrected for multiple (10) comparisons. As noted, there is no significant difference in miR-15b-5p expression in the discovery set and proper statistical evaluation in Figure 5 also removes any putative difference in expression in the replication cohort. Hence, the use of miR-15b-5p expression as a putative diagnostic marker for ASD has not been demonstrated. Greatly increasing the numbers of ASD subjects and controlling for clinical homogeneity (at least regarding intellectual capacity) may yield a fruitful diagnostic marker.
Author Response
We do appreciate the additional crucial comments for the revised manuscript. As the below, responses were made one by one. Hopefully, these responses are sufficient for the reviewer’s comments. The modified points are turned into red words.
Response to Reviewer 1
Minor point 1: There is a typo in Table 1 regarding the age of control subjects.
Thank you for pointing out our typo. We have corrected Table 1.
Minor point 2: In Figures 1 and 5, the correct y-axis label for C) should read miR-19b-3p to be in agreement with the figure legends.
According to your pointing out, we have corrected typos in Figure1 and Figure5.
Major issues 1:
There is a widespread range in miR-15b-5p expression in the ASD subjects in Figure 1A. Examining the discovery ASD cohort as outlined in the Horiuchi et al. (2021) manuscript reveals significant clinical divergence with subjects 3-6 having significantly lower IQs, and therefore lower level of overall functioning, than subjects 1-2. Subjects 3-6 are profoundly impaired, whereas the first two are average. This significant clinical difference may contribute to the expression variance in Figure 1. Although the authors note the sample size as a limitation, the low number of subjects in both the discovery and replication cohorts make interpretation of the results questionable. Significantly increasing the cohort sizes and correcting for overall clinical functioning may yield data demonstrating statistically significant differences in miR expression between the cohorts. Additionally, the lack of data on severity of ASD symptoms in the replication cohort is another complication that makes interpreting the data difficult. The statistical comparisons involving data on miR expression in Figures 1 and 5 should be corrected for multiple (10) comparisons. As noted, there is no significant difference in miR-15b-5p expression in the discovery set and proper statistical evaluation in Figure 5 also removes any putative difference in expression in the replication cohort. Hence, the use of miR-15b-5p expression as a putative diagnostic marker for ASD has not been demonstrated. Greatly increasing the numbers of ASD subjects and controlling for clinical homogeneity (at least regarding intellectual capacity) may yield a fruitful diagnostic marker.
As your suggestions, our study had a great limitation for the number of ASD subjects. However, so far, we are not able to increase the sample size and think that this study should be a pilot study. Fortunately, we could get the intelligence Quotient (IQ) measure about 14 ASD samples among discovery and replication cohorts (In page 6, line 20). IQ was significantly associated with TGFBR3 (r = -0.602, P = 0.023) and MYBL1 (r= -0.562, P = 0.036) as shown but not with miR-15b-5p (Table S4). Additionally, we have added limitation due to the uncertainty of the results, we specified this study is “a pilot study” in the title (In page 1, line 1).
Reviewer 2 Report
The topic of the article refers to an important issue, which is the search for biomarkers allowing for the early diagnosis of autism spectrum disorders (ASD). Among many etiological factors responsible for the development of this disorder, the share of genetic factors is estimated at 50-60%. In this study, the authors evaluated 10 miRNA and mRNA expressions of target genes in peripheral blood to explore a diagnostic biomarker for ASD. They performed an in vitro experiment on control and ASD subjects using HEK293 in order to investigate whether miR-15b-5p regulates the predicted target 22 genes (TGFBR3 and MYBL1). Analysis of results pointed out that “upregulated miR-15b-5p expression may represent a useful diagnostic biomarker of 292 ASD subjects, and it possibly regulates TGFBR3 mRNA expression”. This result is so promising that it may indicate a new approach in research on the pathogenesis of ASD. In terms of methodology, I believe that the research was correct. Also, the analysis of the obtained results and their discussion was carried out correctly. However, taking into account the relatively small number of subjects, I suggest that the authors should indicate in the methodological section that the study is a pilot study. In addition, I suggest the authors refer to the possibility of conducting this type of research in relation to infants and, possibly, in the prenatal period.
Author Response
We do appreciate the additional crucial comments for the revised manuscript. As the below, responses were made one by one. Hopefully, these responses are sufficient for the reviewer’s comments. The modified points are turned into red words.
Response to Reviewer 2
However, taking into account the relatively small number of subjects, I suggest that the authors should indicate in the methodological section that the study is a pilot study.
We agree with your comment. We deleted to describe “a pilot study” in the title (In page 1, line 1) and method section (In page 4, line 10).
In addition, I suggest the authors refer to the possibility of conducting this type of research in relation to infants and, possibly, in the prenatal period.
As per the reviewer’s suggestion, we have mentioned the possibility of biomarkers even in infants and the prenatal period of ASD subjects (In page 9, line 8).
We hope that these responses are suitable for each comment. If you have any questions or comments, please do not hesitate to contact me.
Sincerely,
Jun-ichi Iga, M.D., Ph.D.
Round 2
Reviewer 1 Report
The authors should note that the miR expression results of Figures 1 and 5 are not corrected for multiple (10) comparisons. The statistical analysis section (2.7) should also note attempting to correct for multiple comparisons. The results and discussion sections should note this, and it should be included in the section on limitations.
Author Response
Thank you for your kind suggestion.
